# Biosynthesis of *N-*Docosahexanoylethanolamine from Unesterified Docosahexaenoic Acid and Docosahexaenoyl-Lysophosphatidylcholine in Neuronal Cells

**DOI:** 10.3390/ijms21228768

**Published:** 2020-11-20

**Authors:** Karl Kevala, Michel Lagarde, Arthur A. Spector, Hee-Yong Kim

**Affiliations:** 1Laboratory of Molecular Signaling, National Institute of Alcohol Abuse and Alcoholism, National Institutes of Health, Bethesda, MD 20892, USA; Karl.Kevala@nih.gov (K.K.); spectora@mail.nih.gov (A.A.S.); 2CarMeN Laboratory, INSA-Lyon, 69100 Villeurbanne, France; michel.lagarde@insa-lyon.fr

**Keywords:** synaptamide, docosahexaenoic acid, lysophosphatidylcholine, *N-*docosahexaenoyl phosphatidylethanolamine, plasmalogens, *N-*docosahexaenoyl phosphatidylethanolamine plasmalogen, *N-*acyl phosphatidylethanolamine-phospholipase D, *N-*acylethanolamines, hexachlorophene, bithionol

## Abstract

We investigated the synthesis of *N-*docosahexaenoylethanolamine (synaptamide) in neuronal cells from unesterified docosahexaenoic acid (DHA) or DHA-lysophosphatidylcholine (DHA-lysoPC), the two major lipid forms that deliver DHA to the brain, in order to understand the formation of this neurotrophic and neuroprotective metabolite of DHA in the brain. Both substrates were taken up in Neuro2A cells and metabolized to *N-*docosahexaenoylphosphatidylethanolamine (NDoPE) and synaptamide in a time- and concentration-dependent manner, but unesterified DHA was 1.5 to 2.4 times more effective than DHA-lysoPC at equimolar concentrations. The plasmalogen NDoPE (pNDoPE) amounted more than 80% of NDoPE produced from DHA or DHA-lysoPC, with 16-carbon-pNDoPE being the most abundant species. Inhibition of *N-*acylphosphatidylethanolamine-phospholipase D (NAPE-PLD) by hexachlorophene or bithionol significantly decreased the synaptamide production, indicating that synaptamide synthesis is mediated at least in part via NDoPE hydrolysis. NDoPE formation occurred much more rapidly than synaptamide production, indicating a precursor–product relationship. Although NDoPE is an intermediate for synaptamide biosynthesis, only about 1% of newly synthesized NDoPE was converted to synaptamide, possibly suggesting additional biological function of NDoPE, particularly for pNDoPE, which is the major form of NDoPE produced.

## 1. Introduction

*N-*docosahexaenoylethanolamine (synaptamide) is an endocannabinoid-like metabolite of docosahexaenoic acid (DHA) that is synthesized in the brain [1]. Synaptamide promotes neurogenesis, neurite growth, and synaptogenesis [2,3]. It also attenuates the lipopolysaccharide-induced neuroinflammatory response [4]. These effects are mediated by a cAMP (cyclic adenosine monophosphate)/protein kinase A-dependent pathway that is activated by synaptamide binding to GPR110 (ADGRF1), a Gs_α_ protein-coupled receptor expressed in neural stem cells and the developing brain [5]. This pathway modulates the expression of neurogenic, synaptogenic, and proinflammatory genes and may be a novel target for neurodevelopmental and neuroprotective control [1].

Synaptamide is a structural analog of *N-*arachidonoylethanolamine (anandamide), the potent endocannabinoid synthesized in the brain from arachidonic acid [6,7]. Although the initial studies demonstrated that anandamide is synthesized by direct condensation of arachidonic acid and ethanolamine [8,9,10], subsequent studies indicated that this is unlikely to occur under physiological conditions [11]. The predominant mechanism of anandamide production in the brain is now considered to be the *N-*acylation phosphodiesterase pathway [12,13,14,15,16,17]. This involves the addition of arachidonate to the ethanolamine moiety of phosphatidylethanolamine (PE), forming an *N-*acylphosphatidylethanolamine (NAPE) intermediate, followed by hydrolysis of the *N-*acylethanolamine (NAE) group to form anandamide. The NAPE that is formed is present as both diacyl and plasmalogen forms [18,19,20,21].

The NAPE analogue, *N-*docosahexaenoylphosphatidylethanolamine (NDoPE), has been detected in bovine retina and rabbit cornea, suggesting that like anandamide, synaptamide is synthesized from this intermediate [22,23]. In this study, we have investigated synaptamide formation from DHA in neuronal (Neuro2A) cells to determine whether NDoPE is indeed an intermediate in synaptamide synthesis in the brain. Because DHA can be delivered to the brain either as the free fatty acid or as DHA-lysophosphatidylcholine (DHA-lysoPC) [24,25,26,27], the production of NDoPE and synaptamide was compared when Neuro2A cells were incubated with either unesterified DHA or DHA-lysoPC.

## 2. Results

### 2.1. Time-Dependent NDoPE and Synaptamide Synthesis

Comparative data for DHA incorporation into NDoPE and conversion to synaptamide by Neuro2A cultures during a 24 h incubation is shown in Figure 1. The cells were incubated with a mixture containing 2 μM unesterified [^13^C]DHA and 2 μM DHA-lysoPC containing 87% *sn-*1-DHA-lysoPC, in the presence of the fatty acid amide hydrolase (FAAH) inhibitor URB597. A linear increase in [^13^C]DHA incorporation into *N*-docosahexaenoyl moiety of NDoPE occurred during the first 8 h, but no further appreciable increase was observed thereafter. Although the time-dependent increase of NDoPE production from DHA-lysoPC continued over the entire 24 h incubation, unesterified [^13^C]DHA produced 1.3 to 5.8 times more NDoPE than DHA-lysoPC (Figure 1A).

Mass spectrometric analysis of the NDoPE demonstrated that 75 to 90% of the total DHA incorporated from both substrates was present in plasmalogen-NDoPE (pNDoPE) (Figure 1B) in comparison with diacyl-NDoPE (Figure 1C). For example, at 8 h, 4.6 times more DHA derived from unesterified DHA was incorporated into pNDoPE than into NDoPE. Further analysis of the pNDoPE indicated that more DHA from both DHA substrates was incorporated into the 16:0-plasmalogen NDoPE species (p16:0-NDoPE) than either p18-NDoPE or p18:1-NDoPE at each time point (Figure 1D).

Synaptamide production was measured in the incubation medium [2,3,28]. The amount produced by the Neuro2A cells from unesterified [^13^C]DHA increased linearly during the first 8 h of incubation and then more slowly between 8 and 24 h, whereas synaptamide production from DHA-lysoPC steadily increased throughout the 24 h incubation (Figure 1E). Between 4 to 24 h, 1.7 to 4.2 times more synaptamide was synthesized from unesterified [^13^C]DHA than from DHA-lysoPC.

Only a minute fraction of the DHA incorporated into NDoPE was converted to synaptamide in Neuro 2A cells. After 4 h, the medium contained 26 fmol synaptamide produced from unesterified [^13^C]DHA and 8 fmol produced from DHA-lysoPC. Approximately 3400 fmol of NDoPE was produced from unesterified [^13^C]DHA and 600 fmol from DHA-lysoPC, indicating that synaptamide production accounted for only 0.76–1.3% of the DHA incorporated into total NDoPE. A similar result was obtained after 8 h; synaptamide production accounted for only 1.2–2.7% of the unesterified DHA incorporated into NDoPE and 0.80–0.91% of the DHA from DHA-lysoPC incorporated into NDoPE. Even after 24 h of incubation, synaptamide production accounted for only 1.4–3.3% and 0.80–2.0% of the DHA from unesterified DHA and DHA-lysoPC incorporated into NDoPE, respectively.

### 2.2. Concentration-Dependent NDoPE and Synaptamide Synthesis

Figure 2 shows the concentration dependence of DHA incorporation into total NDoPE, pNDoPE, diacyl NDoPE, and synaptamide. The Neuro2A cells were incubated separately with increasing amounts of [^13^C]DHA or [^13^C]DHA-lysoPC for 8 h at 37 °C. At each substrate concentration, substantially more DHA from unesterified [^13^C]DHA than [^13^C]DHA-lysoPC was incorporated into total NDoPE (Figure 2A), pNDoPE (Figure 2B), and diacyl NDoPE (Figure 2C). At each substrate concentration, pNDoPE accounted for about 80% of the total NDoPE derived from unesterified [^13^C]DHA or [^13^C]DHA-lysoPC. Among pNDoPE species, more DHA from each substrate was incorporated into the p16:0-NDoPE than either p18:0-NDoPE or p18:1-NDoPE (Figure 2D). Increasing amounts of synaptamide were also produced as the concentration of unesterified [^13^C]DHA or [^13^C]DHA-lysoPC was raised (Figure 2E), but from 1.5 to 2.4 times more synaptamide was produced from unesterified [^13^C]DHA. Consistent with the results in Figure 1, the amount of synaptamide produced was only 0.67–1.1% of the DHA incorporated into total NDoPE in the incubations with unesterified [^13^C]DHA, and 0.58–1.8% in the incubations with [^13^C] DHA-lysoPC.

### 2.3. Uptake of Unesterified DHA and DHA-lysoPC

Figure 3 compares the total uptake of DHA in 8 h by Neuro2A cells incubated with increasing amounts of either unesterified DHA or DHA-lysoPC, as measured by gas–liquid chromatography (GC). Increasing amounts of DHA were taken up from both substrates over this range of concentrations. However, at 2, 4, and 8 μM, the DHA uptake from unesterified DHA was 55, 65, and 90% greater than the amount taken up from DHA-lysoPC.

On the basis of these comparative uptake results, we recalculated production of NDoPEs and synaptamide from unesterified DHA and DHA-lysoPC to account for the differential cellular incorporation of DHA from these precursors. Figure 4 shows that the production of total NDoPE (Figure 4A), pNDoPE (Figure 4B), diacyl NDoPE (Figure 4C), NDoPE molecular species (Figure 4D), and synaptamide (Figure 4E) from unesterified DHA and DHA-lysoPC was similar when normalized to account for the differences in DHA uptake. Statistical analysis after normalization indicated that only NDoPE production at 1 and 2 μM showed significant differences between these two substrates, while no significance was found for synaptamide production. Normalization did not alter the higher production of pNDoPE in comparison diacyl-NDoPE, and p16:0-NDoPE still remained the most abundant NDoPE species.

### 2.4. Effect of NAPE-PLD Inhibition on Synaptamide Production

The cytotoxicity of *N-*acylphosphatidylethanolamine-phospholipase D (NAPE-PLD) inhibition was assessed in Neuro2A cultures. Previous studies indicated that hexachlorophene and bithionol produced only small decreases in HEK293 cell viability at concentrations of 8 and 10.5 μM, respectively [29]. On the basis of this information, we tested the effect of these inhibitors on Neuro2A cells at a concentration of 10 μM. As shown in Figure 5, no cytotoxicity was observed in 1.5 h incubations, but a 10–20% reduction in cell viability occurred at 2.5 h. Therefore, a 2 h incubation time, sufficient to obtain reproducible amounts of synaptamide production (Figure 1), was utilized to test the effect of the inhibitors on synaptamide production.

As seen in Figure 6**,** 10 μM hexachlorophene inhibited the production of [^13^C]synaptamide by 65% in Neuro2A cultures incubated with either 2 or 4 μM [^13^C]DHA in the presence of URB597 (Figure 6A, middle set of bars). Bithionol inhibited [^13^C]synaptamide production to a similar extent (Figure 6B, middle set of bars). The production of synaptamide (left set of bars) and anandamide (right set of bars) derived from endogenous substrates was also reduced by these NAPE-PLD inhibitors.

## 3. Discussion

These findings demonstrate that DHA from either unesterified DHA or DHA-lysoPC can be incorporated into NDoPE and converted to synaptamide by neuronal cells. The incorporation of DHA into NDoPE occurred rapidly prior to the formation of synaptamide, as expected for a precursor–product relationship, and indicates that the rate of PE acylation by DHA to form NDoPE is much higher than the rate of NAPE-PLD hydrolysis to release synaptamide. The amount of synaptamide produced was substantially reduced when NAPE-PLD was inhibited by either hexachlorophene or bithionol. Taken together, these results indicate that synaptamide, like its ω-6 structural analogue anandamide that is synthesized from arachidonic acid [13,14,15,16,17], can be produced in neuronal cells at least in part by NAPE-PLD-mediated hydrolysis of a NAPE intermediate.

Although the brain can take up DHA either in the form unesterified fatty acid or DHA-lysoPC [24,25,26,27], recent evidence suggests that the major source is *sn-*1-DHA-lysoPC [26]. However, the present results demonstrate that neuronal cells take up DHA by 1.5 to 2.4 times more from unesterified DHA than DHA-lysoPC when incubated with equimolar concentrations of these substrates. Greater uptake from unesterified DHA occurred when the substrates were incubated either together (Figure 1) or separately (Figure 2) with the cells, and 87% of the DHA-lysoPC incubated with the cells was in the form of *sn-*1-DHA-lysoPC, the isomer that is reported to be transported into the brain by the Mfsd2a transporter [26]. The findings that unesterified DHA is the more effective substrate for Neuro2A cells suggest that any selectivity for transport of DHA-lysoPC probably occurs at the level of the blood–brain barrier, not neuronal cells. Although more DHA from unesterified DHA than DHA-lysoPC was incorporated into NDoPE and synaptamide by the Neuro2A cells, the amounts were similar when they were normalized for the differences in uptake of these substrates (Figure 3 and Figure 4). This suggests that the increased effectiveness of unesterified DHA for NDoPE and synaptamide production is due to the greater capacity of the neuronal cells to take up DHA. Nevertheless, DHA-lysoPC may still substantially contribute to the brain synaptamide production in vivo, as DHA-lysoPC can cross the blood–brain barrier around 10 times more efficiently than unesterified DHA [24].

The plasma concentration of unesterified DHA and DHA-lysoPC in rats was reported to be 1.1–7.7 [25,30] and 1.35 nmol/mL [30], respectively, indicating that both DHA forms are present in the similar micromolar range. The consensus estimates of unesterified DHA and DHA-lysoPC in human plasma are 1.5 and 0.75 nmol/mL [31], respectively, also in the low micromolar range. It was shown that DHA-lysoPC is more effective in targeting the brain for long-term accumulation, but unesterified DHA is kinetically faster in entering the brain [27]. While Mfsd2a is the reported transporter for DHA-lysoPC, it is uncertain whether DHA or other plasma unesterified fatty acids require a membrane transporter to be taken up by the brain. There is evidence indicating that uptake occurs by a diffusion mechanism that does not rely on a membrane protein transporter [32,33,34]. Other findings indicate that a membrane transporter and cytosolic fatty acid-binding protein are required for fatty acid uptake [35,36]. An integrated mechanism that involves diffusion across the lipid bilayer combined with targeting and desorption by binding proteins has been proposed recently [37]. However, further studies will be necessary to resolve this uncertainty regarding the mechanism of plasma unesterified DHA uptake by the brain.

The Neuro2A cells incorporated four times more DHA from either unesterified DHA or DHA-lysoPC into *N*-acyl moiety of pNDoPE than diacyl-NDoPE. This suggests that pNDoPE may be an important source of synaptamide synthesis in neuronal cells and possibly also in the brain. NAPE plasmalogens were detected initially in canine infarcted myocardium [38], and subsequent studies demonstrated their synthesis in homogenates of canine heart [19], canine brain [39], and rat brain [20]. NAPE plasmalogens also are present in fish brain and spinal cord [40], and they are synthesized by rat and dog brain homogenates [20,39]. The *N-*acyl moiety of the fish brain NAPE plasmalogens contains 0.8% DHA [40], and molecular species containing *N-*docosahexaenoyl moiety are present in the NAPE plasmalogens synthesized by COS-7 cells [21]. These results are consistent with the major presence of pNDoPE observed in this study, where exogenous sources of DHA are provided to Neuro 2A cells. In addition to the probable role as an intermediate for synaptamide synthesis, pNDoPE may have additional functions in neuronal cells as only less than 2% of pNDoPE is converted to synaptamide.

The finding that synaptamide production is decreased by NAPE-PLD inhibitors is consistent with the report that *N-*acylethanolamine production is inhibited in the brain of NAPE-PLD gene-deleted mice [41]. Due to cytotoxicity considerations (Figure 5), the concentrations of hexachlorophene or bithionol was limited to 10 μM in the Neuro2A cell incubations. This probably accounts for the fact that the observed reductions in synaptamide production were only 50–60%. However, there are other mechanisms for the production of NAE from NAPE, such as conversion of NAPE to lyso-NAPE or *N-*acyl glycerylphosphorylethanolamine (GPE) prior to the phosphodiesterase-mediated hydrolysis of the NAE moiety [42,43]. In addition, NAE can be produced from NAPE by a phospholipase C-mediated pathway followed by dephosphorylation [44]. Accordingly, it is possible that one or more of these processes may mediate the conversion of the NDoPE to synaptamide, accounting for the incomplete inhibition obtained with the NAPE-PLD inhibitors.

Neuro2A cells also produced unlabeled synaptamide and anandamide during these incubations (Figure 6), indicating that neuronal cells can utilize inherent endogenous substrate stores in addition to the exogenously added [^13^C]DHA. The NAPE-PLD inhibitors significantly decreased the production of these endogenously derived NAEs, suggesting that they also may be formed via a NAPE intermediate. The considerable amount of synaptamide produced from exogenous DHA suggests that inherent synaptamide production in the brain may be increased by factors that stimulate DHA mobilization from neural lipid stores.

In summary, our findings demonstrate that synaptamide production is at least in part mediated through the hydrolysis of NDoPE by NAPE-PLD. Both NDoPE and synaptamide are synthesized in neuronal cells from unesterified DHA and 1-DHA-lyso-PC, two major forms for DHA delivery into the brain. Unesterified DHA is significantly more effective than 1-DHA-lyso-PC in the production of both NDoPE and synaptamide, although the difference is largely due to differences in incorporation efficiency of DHA from these substrates in neuronal cells. Accordingly, any factors that mobilize DHA from neural lipid stores may effectively raise the endogenous synaptamide level in the brain. While the plasmalogen species is the predominant form of NDoPE produced from both substrates, only a minute proportion of NDoPE is converted to synaptamide, suggesting additional function of NDoPE, particularly pNDoPE, other than the intermediary role in synaptamide synthesis.

## 4. Materials and Methods

### 4.1. Substrate Lipids and Internal Standards

Uniformly [^13^C]-labeled DHA ([^13^C]DHA) was a gift from Dr. Anthony Windust (National Research Council, Ottawa, Ontario, Canada), and we synthesized unlabeled and uniformly [^13^C]-labeled DHA-lysoPC. Unlabeled DHA-lysoPC was made by lipolysis of 1-palmitoyl,2-docosahexaenoyl-glycerophosphorylcholine (16:0,22:6-PC) from Avanti Polar Lipids (Alabaster, AL, USA). Briefly, 16:0,22:6-PC in ethylacetate–water (50:50) was incubated with the immobilized lipozyme (Novo Nordisk, Bagsvaerd, Denmark) in the dark at room temperature under nitrogen for 24 h. After centrifugation of the immobilized enzyme, and supernatant concentration, DHA-lysoPC was separated from the remaining 16:0,22:6-PC by thin-layer chromatography with chloroform–methanol–water 60:30:4 as eluant. DHA-lysoPC was quantified on the basis of DHA-methylester obtained by transmethylation, measured by capillary GC. [^13^C]-labeled DHA-lysoPC was obtained by acylation of glycerophosphorylcholine (GPC) according to a method described to prepare 1-acetyl,2-DHA-GPC [45] without the acetylation step. GPC was dissolved in dimethylformamide, and added to 1,3- dicyclohexylcarbodiimide, 4-dimethylamino-pyridine, and [^13^C]DHA for incubation in the dark at room temperature under nitrogen for 24 h. The resulting [^13^C]DHA-lysoPC was purified by thin-layer chromatography and measured according to its DHA content by GC, as stated above for unlabeled DHA-lysoPC.

Diacyl and plasmalogen *N*-17:1 (number of carbons/number of double bonds)-NAPE were synthesized and purified as internal standards for NDoPE on the basis of procedures described in the literature [46]. Briefly, for diacyl species, 18.8 μL (13.3 mM) 15:0,d7-18:1-PE (Avanti Polar Lipids, Alabaster, AL) in dichloromethane (DCM; Sigma-Aldrich, St. Louis, MO, USA) was added to 1.5 mL DCM. A total of 25 μL of pyridine (Sigma-Aldrich) diluted in DCM (1:25 ratio) was added, followed by 62.5 μL 100 mM 4-dimethylaminopyridine (Sigma-Aldrich) and 65 μL 100 mM 17:1 acid chloride (NuChek Prep, Elysian, MN, USA), both in DCM. The mixture was vortexed and incubated overnight at room temperature. The reaction was quenched with saturated NH_4_CO_3_ (Sigma-Aldrich) and centrifuged for 3 min at 2500 rpm (1294 RCF), and the upper phase was discarded. The organic layer was further washed with water several times, dried under N_2_, suspended in 1 mL methyl-*t*-butyl ether (MTBE; Sigma-Aldrich)/chloroform (CHCl_3_; Thermo Fisher Scientific, Pittsburgh, PA, USA)/acetic acid (Thermo Fisher Scientific) (98:2:0.2), and subjected to solid phase extraction (SPE) (Extract-Clean silica, 1000 mg/1.5 mL, American Chromatography Supplies, Vineland, NJ, USA), similar to the method described in [47]. *N*-17:1-PE was eluted into MTBE/CHCl_3_/methanol (Thermo Fisher Scientific) (50:20:30). The eluant was dried under argon and stored at −80 °C in 2:1 CHCl_3_/methanol containing 50 mg/L *t*-butyl hydroxytoluene (BHT, Sigma-Aldrich). For the synthesis of plasmalogen *N*-17:1-PE, p18:0,20:4-PE (Avanti Polar Lipids) was utilized as substrate.

Deuterated synaptamide internal standard (d4-synaptamide) was synthesized by dropwise addition of 100 μL deuterated ethanolamine (d4-ethanolamine, Cambridge Isotope Laboratories, Tewksbury, MA, USA) into ice-cold DCM solution containing 100 mg docosahexaenoyl chloride (NuChek Prep). The mixture was incubated for 15 min on ice and washed several times with liquid chromatography/mass spectrometry (LC–MS)-grade water (Thermo Fisher Scientific) until the final 2 washes were pH-neutral. The organic layer was dried under N_2_, suspended in methanol, and stored under argon at −80 °C.

The quantity of lipid substrates, *N*-17:1-PE, and d4-synaptamide internal standards was calibrated by GC after transmethylation. The positional (1- versus 2-DHA-lysoPC) as well as ^13^C isotopic distribution of DHA-lysoPC and unesterified DHA was characterized by LC–high-resolution MS in both positive and negative mode, using LC conditions described for the synaptamide analysis (Figure 7). Approximately 87% of DHA-lysoPC was 1-DHA-lysoPC (Figure 7A), and 68% of [^13^C]DHA was the fully labeled isotope, [^13^C_22_]DHA (Figure 7B).

### 4.2. Cell Culture and Incubation with Lipid Substrates

Mouse neuroblastoma Neuro2A cells (ATCC, Manassas, VA, USA) were grown in glutamine-free, low glucose Dulbecco’s minimum essential medium (DMEM; Sigma-Aldrich) containing 5% fetal bovine serum (FBS; Sigma-Aldrich) and penicillin/streptomycin antibiotic (Gibco, Gaithersburg, MD, USA) at 37 °C in a 5% CO_2_ atmosphere for up to 4 passages. Prior to treatment with lipid substrate(s), approximately 2.5 × 10^4^ cells/cm^2^ were seeded into 6-well plates or 10 cm cell culture dishes, and after 24 h, the cultures were transferred to 0.25% FBS media. A FAAH inhibitor URB597 (Sigma-Aldrich) was included to prevent synaptamide hydrolysis [2,3,28].

The lipid substrate for the time-dependent studies was a mixture of unlabeled DHA-lysoPC and [^13^C]DHA, while the concentration-dependent studies used each [^13^C]-labeled substrate separately. Dimethyl sulfoxide (DMSO; Sigma-Aldrich) stocks of the URB597 fatty acid amide hydrolase (FAAH) inhibitor and vitamin E (Sigma-Aldrich) containing appropriate lipid substrates were prepared. At 24 h after transfer to the 0.25% FBS medium, DMSO mixture was added to 0.25% FBS in DMEM, such that final media contained 0.1% DMSO, 2 μM URB597, 40 μM vitamin E, and 2 μM of each lipid substrate for time-dependence studies. For the concentration-dependence studies, each labeled substrate at 1–8 μM was incubated for 8 h. At the end of incubation, the media was collected and BHT-methanol was added to make a 30% aqueous solution. Cells were washed twice with phosphate-buffered saline (PBS; Gibco). Both media and cells plates, with 0.8 mL PBS added per well, were stored at −80 °C until processing.

### 4.3. Deacylated NDoPE Analysis

Cells were scraped into glass tubes and suspended in PBS/BHT-methanol/chloroform (0.8:2:1) with internal standards including a diacyl and plasmalogen *N*-17:1-NAPE (15:0,d7-18:1 and p18:0,20:4 species), deuterated phospholipid species (d35-18:0,18:1-PE, p18:0,d9-18:1-PE, d35-18:0-lysoPC (Avanti Polar Lipids)), and 23:0 (Nu-Chek Prep). Lipids were extracted according to the method of Bligh and Dyer [48]. Depletion of phospholipids from 90% of the Bligh–Dyer extracts was achieved by silica SPE using the same solvent system described for *N*-acyl 17:1 NAPE synthesis [38]. The SPE eluant was dried and chemically deacylated in 240 μL of methylamine (40% aqueous, Sigma-Aldrich)/BHT-methanol/1-butanol (Sigma-Aldrich) (4:4:1) mixture at 53 °C [39] for 90 min. The product was dried under N_2_, resuspended in 1 mL of BHT-methanol/water containing 0.1% acetic acid (Thermo Fisher Scientific) (7:3), and loaded onto strata-X polymeric reverse-phase SPE cartridges (Phenomenex, Torrance, CA, USA) pre-equilibrated with water. After washing with 1 mL of the loading solvent followed by 3 mL water (neutral), deacylated NDoPE species were eluted with 3 mL of BHT-methanol, dried under N_2_, and suspended in approximately 17 μL of BHT-methanol for LC–MS/MS analysis (Figure 8A).

Deacylated NDoPE samples were injected onto a Gemini-NX C18 LC column (150 mm × 2.1 mm, 5 μ, Phenomenex), which was coupled to Q-Exactive mass spectrometer to detect the four resulting classes of deacylated compounds, *N*-docosahexaenoyl-glycerylphosphatidylethanolamine (NDoGPE), 1-p16:0-N-DHA-GPE, 1-p18:0-NDoGPE, 1-p18:1-NDoGPE, and 1,2-lyso-*N*-DHA-GPE. LC separation was achieved utilizing a 2-solvent system at 0.4 mL/min flow rate, as described previously [49]: 0.1% NH_4_OH (Fisher Scientific) in solvent A (88%/12% methanol/water) and B (88%/12% methanol/n-hexane (Fisher Scientific)). The gradient consisted of 100% A for 3 min, followed by a linear change to 100% B over 18 min. The column was held at 100% B for an additional 6 min. Between each sample, the column was reconditioned with 0.1 M ammonium acetate (Sigma-Aldrich) for 3 min followed by 100% A for 3 min. The mass spectrometer was run in positive MS/MS mode, and quantitation of each deacylated NDoPE class was achieved via the fragment ions [39] arising from transitions 526.3 to 354.279 (NDoGPE), 548.4 to 376.353 ([13C]NDoGPE), 748.5 to 354.279 (1-p16:0-NDoGPE), 770.6 to 376.353 (1-p16:0-[13C]NDoGPE), 776.6 to 354.279 (1-p18:0-NDoGPE), 798.6 to 376.353 (1-p18:0-[13C]NDoGPE), 774.5 to 354.279 (1-p18:1-NDoGPE), and 796.6 to 376.353 (1-p18:1-[13C]NDoGPE). Peak areas of these transition fragments were compared to those from the N-17:1-PE internal standards: 466.3 to 294.279 (N-17:1-GPE) and 716.6 to 294.279 (1-p18:0-N-17:1-GPE) for diacyl and plasmalogen species, respectively.

### 4.4. Synaptamide Analysis

The procedure for synaptamide analysis is illustrated in Figure 8B. A mixture of deuterated internal standards including d_4_-synaptamide, d_5_-DHA, and d_8_-AA (deuterated free fatty acids, Cayman Chemical, Ann Arbor, MI, USA) was added to media samples, which were made 7:3 BHT-methanol/water and centrifuged for 20 min at 4 °C. Supernatants were loaded onto Strata-X polymeric C18 reverse-phase SPE cartridges (33 μ, 30 mg/mL, Phenomenex) that were wetted with BHT-methanol and equilibrated with water. After being washed with water, samples were eluted with 2.5 mL BHT-methanol into glass tubes, dried under N_2_, and suspended in 25 μL BHT-methanol. An Eclipse C18 UHPLC column (1.8 μ, 2.1 mm × 50 mm, Agilent Technologies, Santa Clara, CA, USA) was coupled to a high-resolution Thermo Scientific Q-Exactive mass spectrometer for analysis. A 3-part tertiary gradient, with all solvents containing 0.01% acetic acid, consisting of water (A), methanol (B), and acetonitrile (Avantor, Radnor Township, PA, USA) (C) was used for LC. After pre-equilibration of column with A/B (60%/40%), 5 μL extract was injected and the solvent composition was linearly changed to A/B/C (36.3%/15%/48.7%) in 5 min, followed by a linear gradient to A/B/C (13.5%/68.4%/18.1%) over 22 min. Positive ion MS/MS was utilized to detect natural, [^13^C]-, and d_4_-synaptamide, using mass transitions of 372.3 to 62.060, 394.4 to 62.060, and 376.3 to 66.085, respectively. The fully labeled [^13^C]synaptamide signal (394.4 to 62.060) was corrected according to the unique isotope distribution for the synaptamide production from the ^13^C-labeled substrates. Quantitation of fatty acid and DHA-lysoPC was achieved by comparison of peak areas of [M + Acetate]^−^ or [M + H]^+^ ions to that of the corresponding internal standard.

### 4.5. Total Lipid Analysis

The remaining 10% of Neuro2A cell Bligh–Dyer lipid extract (see deacylated NDoPE analysis) was transmethylated with boron trifluoride/methanol (Sigma-Aldrich) in the presence of a 23:0 fatty acid internal standard NuChek Prep), and the reaction product was extracted into hexane (Thermo Fisher Scientific). GC analysis was similar to that previously described [49]. The transmethylated samples were injected onto an Agilent 6890 gas chromatograph with a flame ionization detector via a 15 m DB-FFAP phase capillary column (Agilent Technologies). Fatty acid methyl esters were identified according to the elution time and quantified on the basis of the peak area in comparison to the 23:0 internal standard.

### 4.6. NAPE-PLD Inhibition

Effects of the NAPE-PLD inhibitors hexachlorophene and bithionol (Sigma-Aldrich) [29] on cell viability were tested in the presence of 2 μM FAAH inhibitor URB597. Neuro2A cells were seeded at approximately 40% confluency into a 96-well culture plate in 5% FBS/DMEM medium. Then, 24 h later, the media was aspirated and replaced with 0.25% FBS/DMEM containing 2 μM URB597 and 0, 5, or 10 μM hexachlorophene or bithionol, and the incubation was continued for 1.5 or 2.5 h at 37 °C. Cells were lysed and viability determined by an ATP-based luminescence assay (CellTiter-Glo Luminescent Cell Viability Assay Kit, Promega, Madison, WI).

The effect of these NAPE-PLD inhibitors on [^13^C]synaptamide production was also tested in the presence of 2 μM URB597. Neuro2A cells were seeded into 6-well plates (5% FBS/DMEM) and incubated overnight. After changing the medium to 0.25% FBS/DMEM containing 2 μM URB597 and 10 μM hexachlorophene or bithionol and incubating for 15 min at 37 °C, we added [^13^C]DHA to each well along with 40 μM vitamin E, and incubated it for 2 h at 37 °C. Media was collected and d4-synaptamide internal standard was added. Media samples were processed and analyzed as described above. For this assay, endogenous anandamide was also quantitated, utilizing d_4_-synaptamide as an internal standard.

### 4.7. Statistical Analysis

The quantitative results are expressed as means ± standard deviation for triplicate samples, except for viability test of Neuro2A cells incubated with NAPE-PLD inhibitors, where means were based on quadruplicate samples. Statistical analyses were conducted using Student’s *t*-test (Excel software, Microsoft, Redmond, WA, USA). Statistical significance is reported at * *p* < 0.05, ** *p* < 0.01, and *** *p* < 0.001.

## Figures and Tables

**Figure 1 ijms-21-08768-f001:**
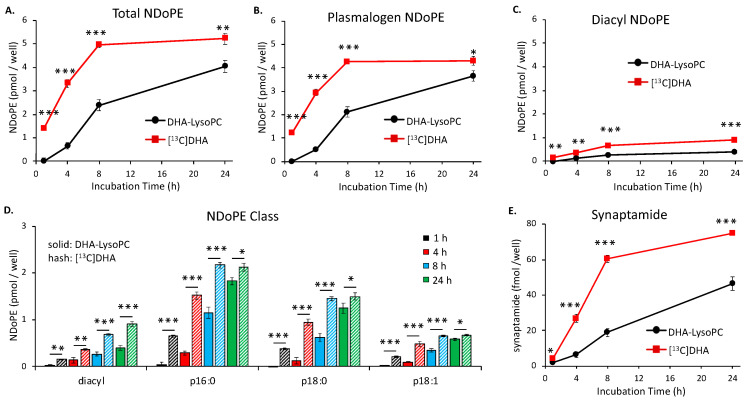
Time-dependent production of *N-*docosahexaenoylphosphatidylethanolamine (NDoPE) and synaptamide from docosahexaenoic acid (DHA) and DHA-lysophosphatidylcholine (DHA-lysoPC) in Neuro2A cells. The cells were incubated with a mixture containing 2 μM each of DHA-lysoPC and [^13^C]DHA, and 2 μM URB597 at 37 °C, and the production of NDoPE (**A**–**D**) and synaptamide (**E**) from these substrates was comparatively evaluated using LC–MS/MS. The production of total NDoPE (**A**), plasmalogen NDoPE (**B**), and diacyl NDoPE (**C**) from DHA and DHA-lysoPC was determined from cell extract in the presence of diacyl or plasmalogen *N*-17:1-phosphatidylethanolamine (PE) internal standards after methylamine-mediated deacylation. The main plasmalogen NDoPE species (p16:0-, p18:0-, and p18:1-NDoPE) are shown along with diacyl NDoPE (**D**). Synaptamide production from DHA and DHA-lysoPC was determined from the culture medium using d4-synaptamide as an internal standard (**E**). Production of unlabeled synaptamide from DHA-lysoPC was adjusted by background subtraction on the basis of the basal values obtained from the untreated control cultures. The significance of the differential production of these metabolites from DHA and DHA-lysoPC was evaluated by *t*-test. The values are the mean ± SD of three separate cultures. * *p* < 0.05; ** *p* < 0.01; *** *p* < 0.001.

**Figure 2 ijms-21-08768-f002:**
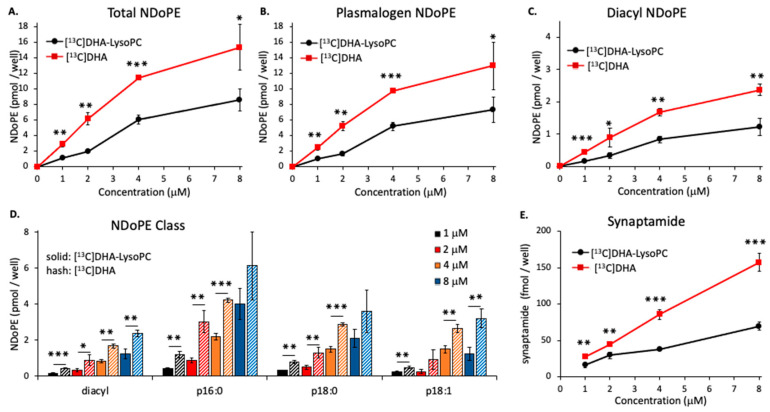
Concentration-dependent production of NDoPE and synaptamide from DHA and DHA-lysoPC. Neuro2A cells were incubated with either [^13^C]DHA-lysoPC or [^13^C]DHA in the presence of 2 μM URB597 for 8 h at 37 °C. The production of total NDoPE (**A**), plasmalogen NDoPE (**B**), and diacyl NDoPE (**C**) from these two substrates were comparatively analyzed after methylamine-mediated deacylation of the cell lipid extract. The main plasmalogen NDoPE species (p16:0-, p18:0-, and p18:1-NDoPE) are shown along with diacyl NDoPE (**D**). The production of synaptamide extracted from culture media is shown in (**E**). The significance of the differential production of these metabolites from DHA and DHA-lysoPC was evaluated by *t*-test. The values are the mean ± SD of three separate cultures. * *p* < 0.05; ** *p* < 0.01; *** *p* < 0.001.

**Figure 3 ijms-21-08768-f003:**
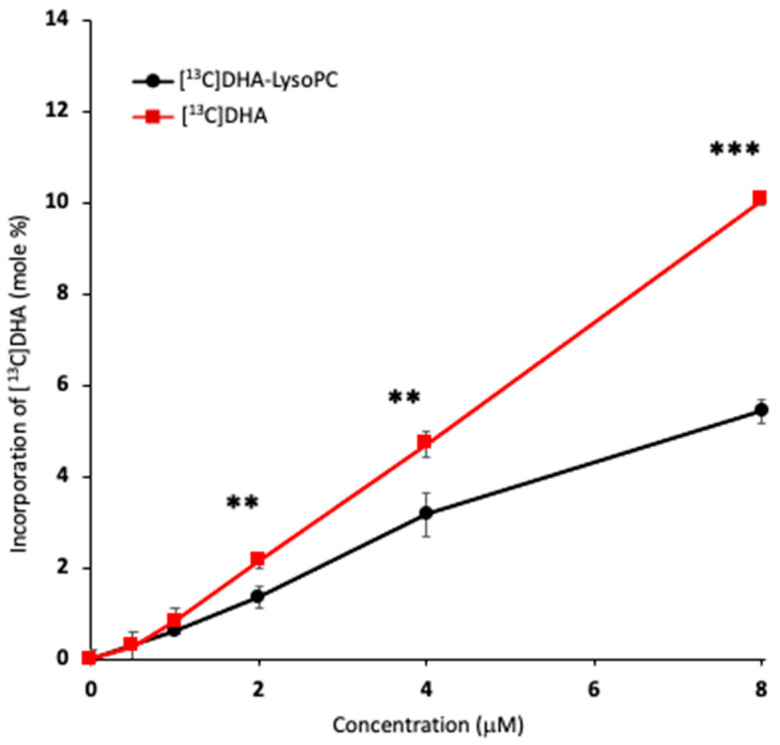
Concentration-dependent DHA incorporation into Neuro2A cells incubated with unesterified DHA or DHA-lysoPC. The cells were incubated with 1–8 μM [^13^C]DHA-lysoPC or [^13^C]DHA for 8 h at 37 °C and the cellular DHA content was analyzed by gas–liquid chromatography (GC) after transmethylation using 23:0 free fatty acid as an internal standard. The total uptake of [^13^C]-DHA by Neuro2A cells after the incubation with unesterified DHA or DHA-lysoPC was compared at equivalent concentrations after subtraction of basal DHA content. Values are the mean ± SD of three separate cultures. ** *p* < 0.01; *** *p* < 0.001.

**Figure 4 ijms-21-08768-f004:**
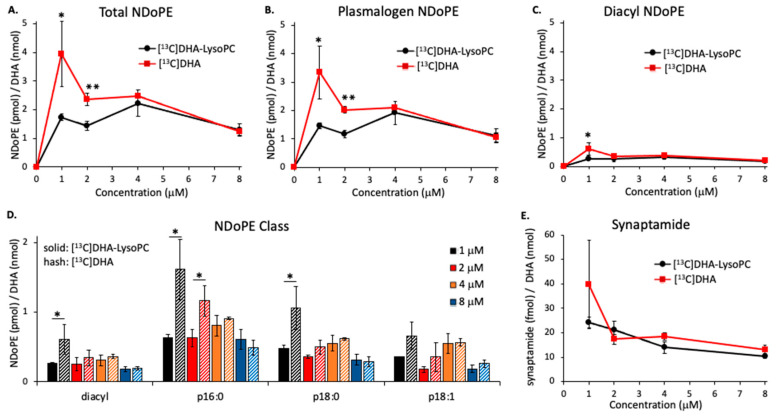
Comparative production profile of total NDoPE (**A**), pNDoPE (**B**), diacyl NDoPE (**C**), NDoPE molecular species (**D**), and synaptamide (**E**) from either [^13^C]-labeled DHA or DHA-lysoPC after normalization for cellular DHA incorporation based on the data shown in Figure 3 * *p* < 0.05; ** *p* < 0.01.

**Figure 5 ijms-21-08768-f005:**
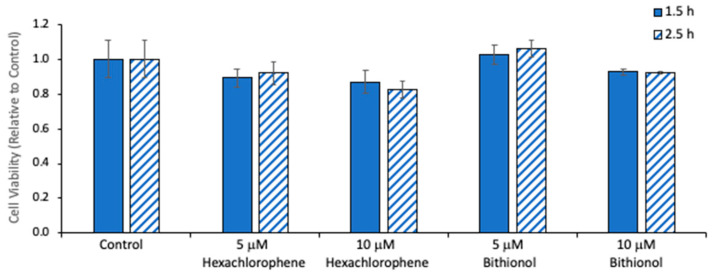
Effect of *N-*acylphosphatidylethanolamine-phospholipase D (NAPE-PLD) inhibitors on cell viability. Viability of Neuro2A cells was determined by ATP assay after incubation with hexachlorophene or bithionol at 5 or 10 μM for 1.5 (solid) or 2.5 h (hash). The values are the mean ± SD of three separate cultures.

**Figure 6 ijms-21-08768-f006:**
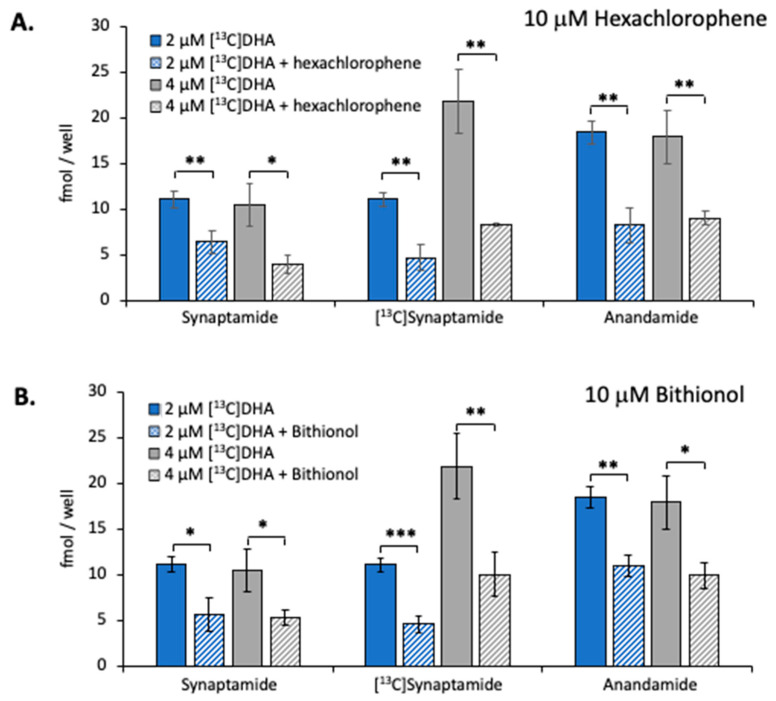
Effect of NAPE-PLD inhibition on synaptamide production. The Neuro2A cells were incubated with 10 μM hexachlorophene (**A**) or 10 μM bithionol (**B**) in the presence of 2 μM URB597 for 2 h with either 2 or 4 μM [^13^C]DHA. The production of [^13^C]synaptmide as well as synaptamide and anandamide from endogenous sources was measured by LC–MS analysis. The values are the mean ± SD of three separate cultures. * *p* < 0.05; ** *p* < 0.01; *** *p* < 0.001.

**Figure 7 ijms-21-08768-f007:**
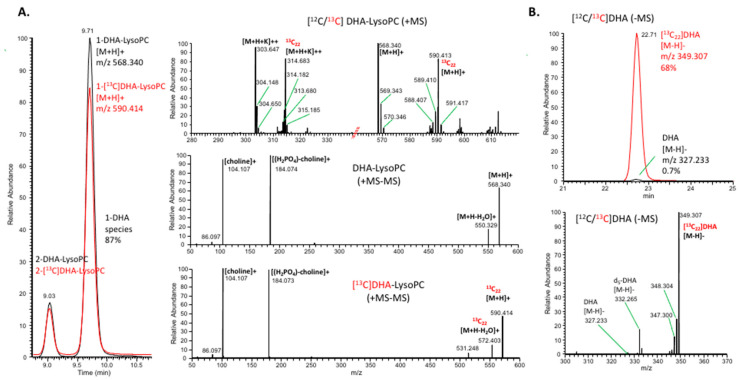
Characterization of [^13^C]DHA-lysoPC and [^13^C]DHA used as substrates. The DHA-lysoPC was analyzed by LC–MS and LC–MS/MS in the positive ion mode (**A**), and [^13^C]DHA by LC–MS in the negative ion mode (**B**). Approximately 87% of labeled or unlabeled DHA-lysoPC was *sn-*1-DHA isomer. Approximately 30% of DHA-lysoPC was fully labeled with [^13^C_22_] (**A**), while approximately 68% of unesterified [^13^C]DHA was fully labeled [^13^C_22_]DHA (**B**). Unlabeled free DHA was present at only 0.7%. The positional and/or isotopic distribution of the substrates was accounted for the quantitative evaluation of labeled products, NDoPEs and synaptamide. A pair of red dotted lines in the X-axis of (**A**) indicates a break in the m/z scale.

**Figure 8 ijms-21-08768-f008:**
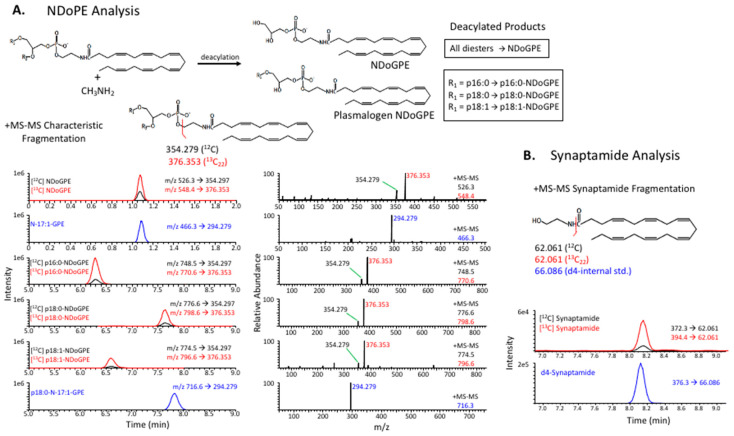
Quantitative analysis of NDoPE and synaptamide by LC–MS/MS. (**A**) Conversion of diverse NDoPE molecular species to four NDoPE-derived species including *N*-docosahexaenoyl-glycerylphosphatidylethanolamine (NDoGPE), 1-p16:0-, 1-p18:0, and 1-p18:1-NDoGPE by methylamine-mediated deacylation, simplifying mass spectrometric analysis on the basis of the characteristic fragments at *m*/*z* 354.279 and *m*/*z* 376.353 formed in the positive ion MS/MS mode (top), and representative ion chromatograms of deacylated NDoPE derived from Neuro2A cells treated with 8 μM [^13^C]DHA and 2 μM fatty acid amide hydrolase (FAAH) inhibitor for 8 h (bottom). The black and red traces correspond to the NDoPE classes derived from endogenous sources and exogenously added [^13^C]DHA, respectively. Blue traces are MS/MS signals for *N*-17:1-GPE and 1-p18:0-*N*-17:1-GPE derived from internal standards (15:0,d7-18:1-*N*-17:1-PE and p18:0,20:4-*N*-17:1-PE, respectively). (**B**) The positive ion MS/MS fragmentation schematic (left) and representative LC–MS/MS ion chromatograms (right) of synaptamide obtained from Neuro2A cell culture media after incubation with 8 μM [^13^C]DHA and 2 μM FAAH inhibitor for 8 h. Both the natural (black trace) and ^13^C-labeled synaptamide (red trace) produced a characteristic fragment at *m*/*z* 62.061 that was used for quantitation against the corresponding fragment from d_4_-synaptamide internal standard at *m/z* 66.086 (blue trace).

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
