# Peer review of "Biosynthesis of N-Docosahexanoylethanolamine from Unesterified Docosahexaenoic Acid and Docosahexaenoyl-Lysophosphatidylcholine in Neuronal Cells"

_ijms, 2020, doi:10.3390/ijms21228768_

Round 1

Reviewer 1 Report

Through binding to GPR110 (ADGRF1) on neural stem cells, N-docosahexaenoylethanolamine (synaptamide) promotes neurodevelopment and attenuates inflammation, indicating it plays an essential role in neuron sciences. However, the synthetic pathways under physiological conditions are still not clear. In the present study, the authors investigated synaptamide synthesis from docosahexaenoic acid (DHA) or DHA-lysophosphatidylcholine (DHA-lysoPC), with forming an intermediate metabolite N-docosahexaenoyl-phosphatidylethanolamine (NDoPE).

Overall, the study is well designed, and the data is well presented. Minor issues are listed below:

  1. The statistical analysis. For triplicate samples, it is incorrect to use Student’s t-test. Nonparametric tests are required.
  2. Please discuss concentrations of DHA and DHA-lysoPC in the plasma so that the significance of Figure 2 can be adjusted. Also, Mfsd2a is the transporter of DHA-lysoPC; what is the transporter of unesterified DHA? Please discuss this issue in the study design.
  3. The font size in the figures are too small, hardly can be seen.

Author Response

  1. The statistical analysis. For triplicate samples, it is incorrect to use Student’s t-test. Nonparametric tests are required.

With all due respect, we disagree with the reviewer’s notion that it is incorrect to use Student’s t-test for triplicate samples. Student’s t-test can be and has been validly used for small n numbers in numerous biochemical studies (de Winter, J.C.F., 2013 "Using the Student's t-test with extremely small sample sizes," Practical Assessment, Research, and Evaluation: 18, Article 10. DOI: https://doi.org/10.7275/e4r6-dj05 https://scholarworks.umass.edu/pare/vol18/iss1/)

  1. Please discuss concentrations of DHA and DHA-lysoPC in the plasma so that the significance of Figure 2 can be adjusted. Also, Mfsd2a is the transporter of DHA-lysoPC; what is the transporter of unesterified DHA? Please discuss this issue in the study design.

As suggested, we have incorporated following paragraph in the manuscript.

The plasma concentration of unesterified DHA and DHA-lysoPC in rats was reported to be 1.1-7.7 (25,30) and 1.35 nmol/mL (30), respectively, indicating that both DHA forms are present in the similar micromolar concentration range.  The consensus estimates of unesterified DHA and DHA-lysoPC in human plasma is 1.5 and 0.75 nmol/mL (31), respectively, also in the low mM range. It was shown that DHA-lysoPC is more effective in targeting the brain for long term accumulation but unesterified DHA is kinetically faster in entering the brain (27). While Mfsd2a is the reported transporter for DHA-lysoPC, it is uncertain whether DHA or other plasma unesterified fatty acids require a membrane transporter to be taken up by the brain. There is evidence indicating that uptake occurs by a diffusion mechanism that does not rely on a membrane protein transporter (32-34). Other findings indicate that a membrane transporter and cytosolic fatty acid binding protein are required for fatty acid uptake (35,36). An integrated view that involves diffusion across the lipid bilayer combined with targeting and desorption by binding proteins has been proposed recently (37). However, further studies will be necessary to resolve this uncertainty regarding the mechanism of plasma unesterified DHA uptake by the brain.

30. Croset, M.; Brossard, N.; Polette, A.; Lagard, M. Characterization of plasma unsaturated lysophosphatidylcholines in human and rat. Biochem.  J. 2000:345,61-67.

31. Bowden J.A. et al. Harmonizing lipidomics: NIST interlaboratory comparison exercise for lipidomics using SRM 1950–Metabolites in Frozen Human Plasma. J. Lipid Res. 2017:58,2276-2288.

  1. Hamilton, J.A.; Johnson, R.A.; Corkey, B.; Kamp, F. Fatty acid transport. The diffusion mechanism in model and biological membranes. Mol. Neurosci. 2001, 16, 99-108.

  2. Jay, A.G.; Simard, J.N.; Huang, N., Hamilton, J.A. SSO and other putative inhibitors of FA transport across membranes by CD36 disrupt intracellular metabolism but do not affect FA translocation. Lipid Res. 2020, 61, 790-807.

  3. Pownall, H.J. Commentary on SSO and other putative inhibitors of FA transport across membranes by CD36 disrupt intracellular metabolism but do not affect FA translocation. Lipid Res. 2020, 61, 595-597.

  4. Glatz, J.F.C.; Luiken, J.J.F.P.; Bonen, A. Involvement of membrane-associated proteins in the acute regulation of cellular fatty acid uptake. Mol. Neurosci. 2001, 16, 123-132.

  5. Veerkamp, J.H.; Zimmerman A.W. Fatty acid binding proteins of nervous tissue. Mol. Neurosci. 2001, 16, 133-142.

  6. Glatz, J.F.C.; Luiken, J.J.F.P. Time for a détente in the war on the mechanism of cellular fatty acid uptake. Lipid Res. 2020, 61, 1300-1303.

  1. The font size in the figures are too small, hardly can be seen.

All figures have been revised using bigger fonts. 

Reviewer 2 Report

The manuscript regards an interesting topic and the authors use appropriate techniques to perform the experiments.
The results are interesting. Unfortunately, the quality and arrangement of the figures are so poor!! Figures, 1, 2, 4, and 8 should be significantly improved. In particular, figures 1F, 2F 4F, and figure 8 (in toto) are not useful for results' comprehension.

Author Response

The manuscript regards an interesting topic and the authors use appropriate techniques to perform the experiments. The results are interesting.

Unfortunately, the quality and arrangement of the figures are so poor!! Figures, 1, 2, 4, and 8 should be significantly improved.

All figures including Figures 1,2,4 and 8 have been revised for clearer presentation.  

In particular, figures 1F, 2F 4F, and figure 8 (in toto) are not useful for results' comprehension

We have removed statistical analysis table in those figures, and the significance is now indicated directly on the graphs with asterisks. Figure 8 has been rearranged for better presentation.